# C-Disentanglement: Discovering Causally-Independent Generative Factors under an Inductive Bias of Confounder

**Xiaoyu Liu[1], Jiaxin Yuan[2], Bang An[1], Yuancheng Xu[2], Yifan Yang[1], Furong Huang[1]**
[1] Department of Computer Science, [2] Department of Mathematics
University of Maryland, College Park
{xliu1231, jyuan98, bangan, ycxu, yang7832, furongh}@umd.edu

## Abstract

Representation learning assumes that real-world data is generated by a few semantically meaningful generative factors (i.e., sources of variation) and aims to discover them in the latent space. These factors are expected to be causally disentangled, meaning that distinct factors are encoded into separate latent variables, and changes in one factor will not affect the values of the others. Compared to statistical independence, causal disentanglement allows more controllable data generation, improved robustness, and better generalization. However, most existing works assume unconfoundedness (i.e., there are no common causes to the generative factors) in the discovery process, and thus obtain only statistical independence. In this paper, we recognize the importance of modeling confounders in discovering causal generative factors. Unfortunately, such factors are not identifiable without proper inductive bias. We fill the gap by introducing a framework named **C**onfounded-**Disentanglement** (C-Disentanglement), the first framework that explicitly introduces the inductive bias of confounder via labels/knowledge from domain expertise. In addition, we accordingly propose an approach to sufficiently identify the causally-disentangled factors under any inductive bias of the confounder. We conduct extensive experiments on both synthetic and real-world datasets. Our method demonstrates competitive results compared to various SOTA baselines in obtaining causally disentangled features and downstream tasks under domain shifts. code available here

## 1 Introduction

Causally disentangled representation learning methods endeavour to identify and manipulate the underlying explanatory causes of variation (i.e., generative factors) within observational data through obtaining *causally disentangled representations* [24, 8]. Such a representation encodes these factors separately in different random variables in the latent space, where changes in one variable do not causally affect the others. Pursuing causal independence[1] makes it possible to identify the ground truth generative factors that are not statistically independent, such as the color, shape and size in a fruit dataset as shown in Figure 1, which is more realistic and allows more controlled data generation, improved robustness, and better generalization in out-of-distribution problems.

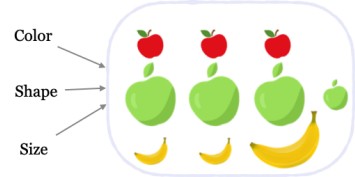

$p(color, shape, size) \neq p(color)p(shape)p(size)$

Figure 1: Causally independent generative factors (color, shape, size) may appear statistically correlated in observational data.

---

[1]We will use causally disentangled and causally independent interchangeably.

37th Conference on Neural Information Processing Systems (NeurIPS 2023).

Despite great success, existing methods suffer from the identifiability issue in discovering semantically meaningful generative factors. The main reason is that most of them equate disentangling in the latent space with enforcing statistical independence [9, 11, 6] in the latent space, or requiring no mutual information[8]. In other words, they explicitly or implicitly assume that the observational dataset is *unconfounded* in the generative process, which is not realistic in practice. A confounder $\mathbf{C}$ is a common cause of multiple variables. It could create a correlation among causally disentangled generative factors in the observational distribution. In the previous example of a fruit dataset, the type of fruit is a confounder, creating a correlation between size and shape (apples are usually red and round objects), thus size and shape can never be captured by a learning objective that requires statistically independent latent representations. Assuming unconfoundedness in such a dataset leads to discrepancies in characterizing the statistical relationship of the generative factors, and makes them *non-identifiable*.

Considering the limitation of the unconfoundedness assumption, a few of the recent studies take into account confounders [24, 23, 21] in the problem formulation. Specifically, [23, 21] proposed evaluation metrics in the confounded causal generative process. [24] introduced an unsupervised learning scheme by regulating the learning process with the Independence-Of-Support Score (IOSS) under an unobserved confounder, following the theory that if a latent representation is causally disentangled, then it must have independent support. However, it has been shown *almost impossible* to obtain disentangled representations through purely unsupervised learning without proper inductive biases [10, 16]. While there are a few semi-supervised or weakly supervised methods [16, 17] developed seeking statistical independence, none of the above-mentioned methods considers how to provide inductive bias for confounder in learning causally disentangled representations.

In this paper, we recognize the importance of providing inductive bias to confounder so that the ground truth generative factors can be identified and we fill the gap by introducing the inductive bias via knowledge from domain expertise. Specifically, we propose a framework called **C**onfounded-**Disentanglement** (*C-Disentanglement*). C-Disentanglement is, to the best of our knowledge, the first framework that discusses the identifiability issue of generative factors regarding the inductive bias of confounder, and thus opens up the possibility to discover the ground truth causally disentangled generative factors which are correlated in the observational dataset. Importantly, C-Disentanglement is a general framework that includes existing methods (in which unconfoundedness is assumed) as special cases, allowing customized inductive bias of confounders according to diverse user needs.

Under the framework, We develop an algorithm to discover the causally disentangled generative factors in the latent space with inductive bias $\mathbf{C}$, where $\mathbf{C}$ is a label set. Instead of enforcing global statistical independence among variables on the observational dataset, we partition the dataset into subsets according to realizations of $\mathbf{C}$, and regulate these variables within each subset.

Our proposed discovery methodology is general and can be applied to various disentangle mechanisms. But as a prototype for a proof of concept, we implement our method, cdVAE, under the context of VAE [12]. Specifically, we formulate a mixture of Gaussian model in which each Gaussian component, conditioned on a specific value of $\mathbf{C}$, represents the distribution of a latent variable that estimates the underlying generative factor. Intuitively, under the previous example of the fruit dataset, if we fix the type of fruit (e.g., the apple) and check the correlation between the color and the shape, we could find that the color change within the apples does not affect the shape distribution, thus can be captured by regulating the statistical independence conditioned on $\mathbf{C}$ (a fixed fruit type).

Our methodology is a novel learning paradigm that can discover latent factors through weak supervision labels (i.e., $\mathbf{C}$).

We experiment with several tasks from image reconstruction and generation, to classification under the out-of-distribution scenario on both real-world and synthetic datasets. We demonstrate that our method outperforms various baselines in discovering generative factors in the latent space.

**Summary of contributions: (1)** We recognize the identifiability issue of discovering generative factors in the latent space. We accordingly introduce a framework, named Confounded Disentanglement (C-Disentanglement). It is the first framework that discusses how inductive bias of confounder could be explicitly provided via labels/knowledge from domain expertise. **(2)** We propose an algorithm, cdVAE, that discovers causally disentangled generative factors in the latent space. The algorithm sheds light on the easy injection of inductive bias into existing methods.**(3)** We conduct extensive experiments and ablation studies across various datasets and tasks. Empirical results verify that

cdVAE outperforms existing methods in terms of inferring causally disentangled latent representation and also show cdVAE's superiority in downstream tasks under OOD generalization.

## 2 Preliminaries

In this section, we introduce basic concepts in causal inference and then show how to evaluate the causal relationship among variables in latent space.

**Causal graph through DAG.** The causal relationship among variables can be reflected by a Directed Acyclic Graph (DAG). Each (potentially high-dimensional) variable is denoted as a node. The directed edges indicate the causal relationships and always point from parents to children.

**Intervention and do-operator.** *Intervention* is one of the fundamental concepts in causal inference. When we intervene on a variable, we set the value of a variable and cut all incoming causal arrows since its value is thereby determined only by the intervention [20]. The intervention is mathematically represented by the *do-operator* $do(\cdot)$. Let $Z_1$ and $Z_2$ be two variables, $P(Z_2|do(Z_1 = z_1))$ characterizes an interventional distribution and reflects how a change of $Z_1$ affects the distribution of $Z_2$. The do-operation and the interventional distribution should be estimated on the interventional dataset. However, in practice, the true distribution of the data is unavailable but an observational subset. As a result, we estimate the interventional distribution from the observational set following *do-calculus* introduced by [19]. A detailed introduction can be found in the Appendix A.

**Confounders bring in spurious correlation.** Although the detailed definitions vary from literature, a confounder usually refers to a common cause (i.e., a common parent in the causal graph) of multiple variables and it brings in a certain level of correlation among these variables. Consequently, to estimate the causal effect of one variable on the other from the observational dataset, we have to eliminate the correlation introduced by the confounders. For example, when analyzing the causal relationship between the number of heat strokes and the rate of ice cream consumption, we may find that there is a correlation. However, the temperature is a confounder in the situation. If we eliminate this spurious correlation by conditioning on the temperature, we may find out that under the same temperature, heat stroke is not correlated to the rate of ice cream consumption.

**Evaluation of interventional distribution.** We specifically consider the case where there are variables $Z_1$ and $Z_2$, and we analyze how the existence of parental nodes affects the estimation of $P(Z_2|do(Z_1 = z_1))$ on the observational distribution.

**Proposition 2.1.** *Let $Z_1$ and $Z_2$ be two random variables, $\mathbf{C}^*$ be the ground truth confounder set. If $\mathbf{C}$ is a superset of or is equivalent to $\mathbf{C}^*$, i.e., $\mathbf{C}^* \subseteq \mathbf{C}$, with $c$ being a realization of $\mathbf{C}$, we have*

$$P(Z_2|do(Z_1)) = \sum_{c \in \mathbf{C}} P(Z_2|Z_1, \mathbf{C} = c)P(\mathbf{C} = c) \tag{1}$$

*if no $C \in \mathbf{C}$ is a descendent of $\mathbf{Z}$.*

The proof can be found in Appendix B.1.

Intuitively speaking, Proposition 2.1 states that to accurately estimate the causal relationship between variables, we have to eliminate the spurious correlation by conditioning on confounders. In addition, conditioning on additional variables will not affect the estimation if they are not decedents of $\mathbf{Z}$.

**Causal disentanglement or statistical independence.** If we had access to a series of ground-truth features that generated a fruit dataset(e.g., color, shape, texture), we can encode each of the features into a variable and concatenate them for a disentangled representation. However, these features in the observational dataset are usually correlated, depending on the type of fruits. If not considered from the causal perspective, traditional methods aiming for statistically independent representations, such as $\beta$-VAE [9], may not be able to consider correlated features that are disentangled. In fact, neither causal disentangled entails statistical independence nor vice versa because of the existence of confounders. We, therefore, formulate the task of inferring causally disentangled features from the observational under a certain level of confoundedness.

# 3 Confounded causal disentanglement via inductive bias

## 3.1 Problem formulation

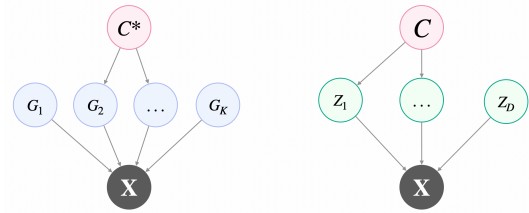

We formally frame the task of discovering a set of generative factors from the observational dataset from a causal perspective as shown in Figure 2. Let $\mathbf{X}$ denote the observational data, the confounded causal generative process [23, 21] assumes that $\mathbf{X}$ are generated from $K$ ground-truth causes of variations $G = [G_1, G_2, ..., G_K]$ (i.e., $G \to X$) that do not cause each other. These generative factors are generally not available, and they are confounded

Figure 2: The left figure shows the ground truth generative process while the right figure demonstrates the learning task.

by some unobserved confounding variables $\mathbf{C}^*$. The task of interest is to discover those generative factors in the latent space (denoted as $\mathbf{Z} = [Z_1, Z_2, ..., Z_D]$) that best approximates $\mathbf{G}$ from $\mathbf{X}$.

**Causal disentanglement among latent generative factors.** Generative factors, encoded in the latent representational space, are causally disentangled, meaning that intervening on the value of one factor does not affect the distribution of the others. It is formally defined as:

**Definition 3.1** (Causal Disentanglement on Data $\mathbf{X}$([19, 23, 24]))**.** A representation is disentangled if, for $i \in \{1, ..., D\}$,

$$P(Z_i | \text{do}(Z_{-i} = z_{-i}), \mathbf{X}) = P(Z_i | \mathbf{X}), \quad \forall z_i. \tag{2}$$

where $-i = \{1, 2, ..., D\}/i$ indicates the set of all indices except for $i$.

**Challenge of unobserved $\mathbf{C}^*$.** The generative factors $\mathbf{G}$ are not identifiable without proper inductive bias of $\mathbf{C}^*$ [16]. Previous works on discovering the generative factors either obtain disentanglement by enforcing statistical independence on latent variables [9, 15, 6], or require that latent variables do not capture information of each other [8]. Such a setting is equivalent to assuming unconfoundedness of the generative factors. It ignores the possibility that correlated latent variables can also be causally disentangled in the observational distribution, and hence is an assumption too restrictive. Fortunately, even though the ground truth generative factors are unobserved, domain expertise may inform a "reasonable" or "likely" inductive bias of the confounder from an accessible label set, denoted as $\mathbf{C}$. This $\mathbf{C}$ is used to account for all correlation among $\mathbf{Z}$.

Note here that we do not assume the accessible label set $\mathbf{C}$ equals to $\mathbf{C}^*$ (but hope that it is close to $\mathbf{C}^*$). The relationship between $\mathbf{C}^*$ and label set $\mathbf{C}$ must fall into one of the following scenarios, despite the immeasurability of their exact relationships.

**Case 1** The label set contains no information about the confounders, i.e., $\mathbf{C} = \emptyset$

**Case 2** The label set contains partial information about the confounders, i.e.,$\mathbf{C} \subset \mathbf{C}^*$,

**Case 3** The label set contains all information about the confounders, i.e.,$\mathbf{C} = \mathbf{C}^*$

One may argue that $\mathbf{C}$ may contain information irrelevant to the ground truth confounders. We show in Appendix B that in the confounded generative process described in this paper, irrelevant information in $\mathbf{C}$ does not affect the evaluation of the interventional distribution, and therefore can be ignored. We only take into consideration how much information in $\mathbf{C}^*$ is captured here without loss of generality.

According to Proposition 2.1, in case 3, we can estimate $P(Z_i | do(Z_{-i} = z_{-i}, \mathbf{X})$ on the observational set with inductive bias from $\mathbf{C}$. However, in the rest of the cases, the equation does not hold. Therefore, we introduce an operator $do^c(\cdot)$ to estimate the interventional distribution on the observational set under inductive $\mathbf{C}$. The framework that applies $do^c(\cdot)$ is named C-Disentanglement, shorten for *Confounded Disentanglement*.

## 3.2 C-Disentanglement as the learning objective

We formally introduce C-Disentanglement and $do^c$ as follows:

**Definition 3.2** (C-Disentanglement and $do^c$)**.** Let $\mathbf{X}$ be the observational data, $\mathbf{Z} = [Z_1, Z_2, ..., Z_D]$ be a concatenation of $D$ random variables, $\mathbf{C}$ be a label set selected from domain expertise to provide inductive bias for confounders of the observational data, we define $do^c$ operation as:

$$P(Z_i|do^c(Z_{-i} = z_{-i}), \mathbf{X}) = \sum_{c \in \mathbf{C}} P(Z_i|\mathbf{X}, Z_{-i} = z_{-i}, \mathbf{C} = c)P(\mathbf{C} = c) \tag{3}$$

$\forall i \in 1, 2, 3, ..., D$, where $c$ is realizations of $\mathbf{C}$. $\mathbf{Z}$ obtains C-Disentanglement on $\mathbf{X}$ given $\mathbf{C}$ if

$$P(Z_i|do^c(Z_{-i} = z_{-i}), \mathbf{X}) = P(Z_i|\mathbf{X}). \tag{4}$$

**C as an approximation of $\mathbf{C}^*$.** Whether the ground truth generative factors can be discovered depends on whether their statistical relationship on the observational distribution is correctly characterized, and it requires knowledge of $\mathbf{C}^*$. However, due to the unobservable nature of the confounders, we use $\mathbf{C}$ as an approximation of $\mathbf{C}^*$, as in Equation (4). Then the identifiability of the ground truth generative factors depends on the relationship between $\mathbf{C}$ and $\mathbf{C}^*$. We thus examine three possibilities as listed in section 3.1 together with a case study as in Figure 3.

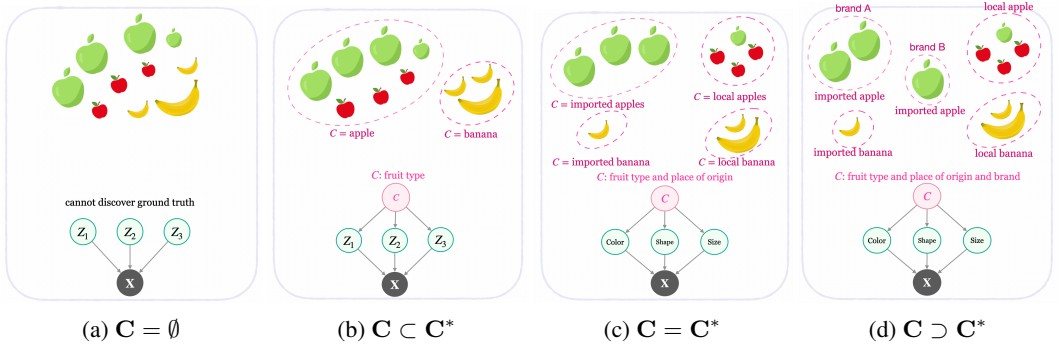

(a) $\mathbf{C} = \emptyset$    (b) $\mathbf{C} \subset \mathbf{C}^*$    (c) $\mathbf{C} = \mathbf{C}^*$    (d) $\mathbf{C} \supset \mathbf{C}^*$

Figure 3: **Case study: a fruit dataset.** Suppose a fruit dataset is generated by *color, shape and size.* These factors are correlated depending on the type of fruit and the place of origin. One way to estimate the causal effect within latent variables, proposed as C-Disentanglement, is to condition on realizations of the confounders $\mathbf{C}$, which partitions the data into subsets and inspects within each set. Figure (a)-(d) demonstrate partitions under different $\mathbf{C}$. In fig. 3a, when $\mathbf{C} = \emptyset$, color, shape and size cannot be identified, as the requirement of statistical independence contradicts the fact that they are correlated in the observational set. In fig. 3b, only partial information is given. It still helps as it can identify that size and shape are causally independent although entangled by color. In fig. 3c and fig. 3d, the ground truth generative factors can be discovered.

In **Case 1**, $\mathbf{C}$ is an empty set as shown in fig. 3a. It is equivalent to assuming that the generative process is unconfounded and Equation (3) degrades to statistical independence on the observational set:

$$P(Z_i|do^c(Z_{-i} = z_{-i}), \mathbf{X}) = P(Z_i|Z_{-i} = z_{-i}, \mathbf{X}) = P(Z_i|\mathbf{X}). \tag{5}$$

This contradicts the fact that these factors are actually correlated and such an assumption makes the ground truth generative factors unidentifiable when there exists a underlying confounder.

In **Case 3**, as shown in fig. 3c and fig. 3d, $\mathbf{C} \supseteq \mathbf{C}^*$, $do$ is equivalent to $do^c$. Thus the ground truth generative factors can be identified.

In **Case 2**, partial information about the confounders is given as shown in fig. 3b. Heuristically speaking, this relaxes the constraint of statistical independence on the observational set and therefore enlarges the solution set, providing an increased possibility for the ground truth factors to be recovered. We empirically show in Section 5 that even partial information of confounders improves accuracy, outperforming existing methods in various tasks, even under distribution shifts.

## 4 cdVAE: identify causally disentangled factors in the latent space.

In this section, we provide an algorithm, cdVAE, for confounded disentangled VAE, to identify the latent causally disentangled generative factors under confounder $\mathbf{C}$ in the context of VAE.

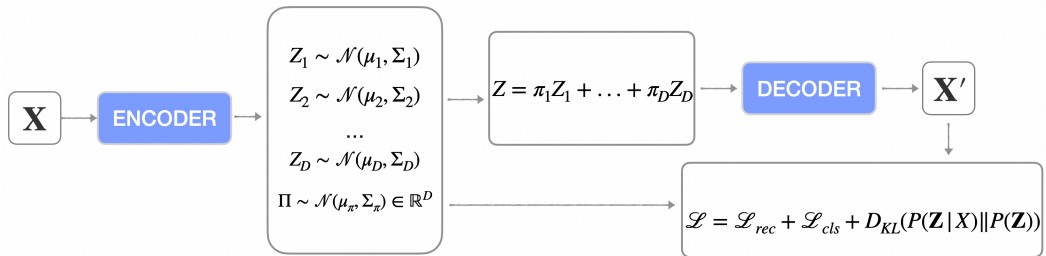

Figure 4: **Learning paradigm of cdVAE.** The input data $\mathbf{X}$ is partitioned by realizations of the confounder $\mathbf{C}$. We infer from each partition a Gaussian distribution and form a mixture of Gaussian model to characterize the distribution of $\mathbf{Z}$ on the observational distribution. We assume soft assignment of samples and further infer $\pi^c(\mathbf{X})$ that resembles $\mathbf{C}$.

## 4.1 Learning objective

Given $\mathbf{X}$ as the dataset, we hope to find a deterministic function $f$, parameterized by $\theta$, where $f : \mathcal{Z} \to \mathcal{X}$ such that (1) $P(X) = \int f(\mathbf{Z}; \theta) P(\mathbf{Z}) dz$ is maximized and (2) each $Z_i$ encodes causally disentangled generative factors, with $\mathbf{Z} = [Z_1, Z_2, ..., Z_D]$ as a concatenation of random variables in the latent space.

More concretely, $f(z; \theta)$ is characterized by a probability distribution $P(\mathbf{Z}|\mathbf{X}; \theta)$, and $p(\mathbf{Z})$ is a prior distribution from where $\mathbf{Z}$ can be easily sampled. For each $Z_i \in \mathbf{Z}$, we require

$$P(Z_i | do^c(Z_{-i}), \mathbf{X}) = P(Z_i|\mathbf{X}), \quad \forall c \in \mathbf{C}. \tag{6}$$

Applying the variational Bayesian method [12], the learning object is to optimize the evidence lower bound (ELBO) while satisfying the constraints on causal disentanglement:

$$\max_{\theta, \phi} \quad \mathbb{E}_{\mathbf{Z} \sim Q}[\log P(\mathbf{X}|\mathbf{Z}; \theta)] - D[Q(\mathbf{Z}|\mathbf{X}; \phi) || P(\mathbf{Z})] \tag{7}$$

$$s.t. \quad P(Z_i | do^c(Z_{-i}), \mathbf{X}) - P(Z_i|\mathbf{X}) = 0 \quad \forall i \in 1, 2, ..., D \tag{8}$$

For simplicity, we omit all model parameters $\theta, \phi$ in writing.

## 4.2 Learning strategy

We start from the estimation of the causal disentanglement constraint as shown in Equation (6). Because the probability distribution is hard to directly calculate, inspired by [23], we resort to its first-order moment as an approximation. Specifically, we estimate the $L_1$ distance between the expectation of probability $P(Z_i | do^c(Z_{-i}), \mathbf{X})$ and $P(Z_i|\mathbf{X})$ as follows:

$$l_c = \sum_{i=1}^{D} d[\mathbb{E}(Z_i | do^c(Z_{-i}), \mathbf{X}), \quad \mathbb{E}(Z_i|\mathbf{X})]. \tag{9}$$

We show in the following theorem that eq. (9) is satisfied the learned latent variable $\mathbf{Z}$ subjects to a Gaussian distribution with a diagonal covariance matrix. Proof can be found in appendix B.2.

**Theorem 4.1.** *Suppose that the latent variable $\mathbf{Z}$ on dataset $\mathbf{X}$ given $\mathbf{C} = c$ subjects to the Gaussian Distribution $\mathcal{N}(\mu^c(\mathbf{X}), \Sigma^c(\mathbf{X}))$. Specifically,*

$$P(\mathbf{Z}|\mathbf{C} = c, \mathbf{X}) = (2\pi)^{-D/2} \det(\Sigma^c)^{-1/2} \exp\left( -\frac{1}{2}(\mathbf{Z} - \mu^c)^{\mathsf{T}}(\Sigma^c)^{-1}(\mathbf{Z} - \mu^c) \right),$$

*where $\mathbf{Z} \in \mathbb{R}^D$. If $\Sigma^c(\mathbf{X})$ is diagonal for all $c$, we have*

$$l_c = \sum_{i=1}^{D} d(\mathbb{E}(Z_i | do^c(Z_{-i}), \mathbf{X}), \quad \mathbb{E}(Z_i|\mathbf{X})) = 0. \tag{10}$$

From Theorem 4.1, we see that for each $\mathbf{C} = c$, enforcing the latent variable $\mathbf{Z}$ to be statistically independent minimizes $l_c$. Taking the whole $\mathbf{C}$ set into consideration, $P(\mathbf{Z}|\mathbf{X})$ subjects to a mixture

of Gaussian distribution where each centroid is inferred from observational data under a specific realization of the confounder $\mathbf{C}$:

$$P(\mathbf{Z}|\mathbf{X}) = \sum_{c \in C} \pi_c \mathcal{N}(\mu^c(\mathbf{X}), \Sigma^c(\mathbf{X})). \tag{11}$$

The mixing coefficient $\pi_c = P(\mathbf{C} = c|\mathbf{X})$ reads the probability of occurrence of $\mathbf{C} = c$. Nevertheless, such a hard assignment of coefficient varies with observable dataset and cannot accommodate scenarios in which the label set $\mathbf{C}$ does not exist. In this paper, we parameterize $\pi_c$ as a Gaussian distribution for a soft assignment of samples: $\pi_c \sim \mathcal{N}(\mu^c(\mathbf{X}), \Sigma^c(\mathbf{X}))$. Parameters $\mu^c(\mathbf{X})$ and $\Sigma^c(\mathbf{X})$ are learned to minimize the discrepancy with $P(\mathbf{C} = c|\mathbf{X})$.

In VAEs, the prior of the latent space $P(\mathbf{Z})$ is assumed to follow a Gaussian distribution with mean zero and identity variance: $\mathbf{Z} \sim \mathcal{N}(\mathbf{0}, I)$. To avoid enforcing statistical independence of the overall latent spaces we learn, we only assume that the prior of $\mathbf{Z}$ to be a Gaussian distribution with variance one for each subset of $\mathbf{C}$.

Concretely, suppose that the latent variable $\mathbf{Z}$ for $\mathbf{X}$ under $\mathbf{C} = c$ follows a Gaussian distribution $\mathcal{N}(\mu^c(\mathbf{X}), \Sigma^c(\mathbf{X}))$, then the KL divergence in (7) regulates the distribution to $\mathcal{N}(\mu^c(\mathbf{X}), I)$.

By the Lagrangian multiplier method, the new loss function is

$$\mathcal{L} = \underbrace{-\mathbb{E}[\log P(\mathbf{X}|\mathbf{Z})]}_{\mathcal{L}_{rec}} + \underbrace{\mathbb{E}[\log P(\mathbf{C}|\pi_c(\mathbf{X}))]}_{\mathcal{L}_{cls}} + D_{KL}[P(\mathbf{Z}|\mathbf{X}, \mathbf{C})||P(\mathbf{Z}|\mathbf{C})]. \tag{12}$$

The algorithm pseudocode can be found in Appendix C.

## 5   Experiments

In this section, we experimentally compare cdVAE with various baselines on synthetic and real-world datasets, and study properties of cdVAE through the ablation studies. We demonstrate that cdVAE allows for causally disentangled but statistically correlated features being discovered in the latent space, which better approximates the ground truth generative factors and outperforms SOTA baseline under distribution shifts.

Concretely, we aim to answer the following questions regarding the proposed model:

▷ **Q1**: How does it perform compared to the existing methods in the latent space?
▷ **Q2**: How does it perform in downstream tasks such as classification under distribution shifts?

### 5.1   Basic setup

**Datasets.** we evaluate cdVAE on three datasets: synthetic datasets 3dshape [3] and Candle [21], and real-world dataset CelebA [15]. 3dshape is a dataset of 3D shapes generated from 6 ground-truth independent latent factors. These factors are floor color, wall color, object color, scale, shape, and orientation. There are 480,000 images from all possible combinations of these factors, and these combinations are present exactly once. Candle is a dataset generated using Blender, a free and open-source 3D CG suite that allows for manipulating the background and adding foreground elements that inherit the natural light of the background. It has floor hue, wall hue, object hue, scale, shape, and orientation as latent factors.

We use 3dshape and Candle as synthetic datasets. To mimic the real-world scenario where perfectly disentangled ground-truth generative factors do not exist or are hard to identify, we manually create a certain level of correlation in these datasets by first making rules among certain attributes and then sampling accordingly.

**Baselines and tasks**. We compare the performance of cdVAE in the task of image generation and classification under distribution shift with the following architectures: (1) VAE-based methods (vanilla VAE [12], $\beta$-VAE [9], FactorVAE [11]) that equate disentanglement as statistical independence, (2) Existing causal regulation methods, CAUSAL-REP [24], (3) cVAE [12] as it also applies the label information (4) GMVAE [7] as it also adopts a mixture of Gaussian model in a variational autoencoder framework.

Table 1: **Compare with baselines in image generation task on celebA and candle.** The reconstruction error indicates the end-to-end performance of the image generation task. D-score measures from a non-causal perspective and requires that the generation process is unconfounded. We expect that a good method that recovers causally disentangled factors should obtain poor (i.e., low) D-scores. IOSS, UC and CG are causal metrics that measure the level of disentanglement of a representation.

| Methods | CelebA | | | Candle | | | | |
|---|---|---|---|---|---|---|---|---|
| | Recon ↓ | D ↑ (non-causal) | IOSS ↓ (causal) | Recon ↓ | D ↑ (non-causal) | UC ↑ (causal) | CG ↑ (causal) | IOSS ↓ (causal) |
| VAE | 0.33 | 0.11 | 0.78 | .024 | 0.14 | 0.10 | 0.18 | 0.69 |
| $\beta$-VAE | 0.27 | 0.15 | 0.74 | .017 | **0.18** | 0.11 | 0.24 | 0.54 |
| FactorVAE | 0.25 | **0.17** | 0.68 | .014 | 0.15 | 0.13 | 0.26 | 0.51 |
| CAUSAL-REP | 0.29 | 0.16 | 0.34 | .012 | **0.18** | 0.20 | 0.32 | 0.31 |
| cVAE | 0.32 | 0.14 | 0.64 | .020 | 0.14 | 0.12 | 0.21 | 0.62 |
| GMVAE | 0.30 | 0.13 | 0.71 | .018 | 0.12 | 0.09 | 0.16 | 0.71 |
| cdVAE | **0.18** | 0.12 | **0.21** | **.008** | 0.11 | **0.35** | **0.54** | **0.16** |

Note that in this paper, we do not compare cdVAE with CausalVAE [27], despite the latter also obtaining "disentanglement" in the latent space. The main reason is that the problem-setting and the learning objectives are fundamentally different. CausaslVAE aims to disentangle known ground truth generative factors in the latent space, it aims to bind these factors one by one to latent variables and the causal dependency is not considered. The task of interest in this paper is to discover causally disentangled factors in the latent space and the ground truth is unavailable, thus these two methods are not comparable.

**Evaluation metrics**. The experimental results are evaluated on (1) end-to-end evaluation metrics: the accuracy of classification under distribution shifts and reconstruction loss in image generation task. (2) how well the learned latent generative factors recover the ground truth one: Maximal Information Coefficient (MIC) and Total Information Coefficient (TIC) [13]. (3) Existing disentanglement scores from a causal perspective: IRS [23], UC/CG [21], IOSS [24], and statistical perspective: D-Score [8]. The higher these evaluation metrics, the better the model except for IOSS (the lower, the better). A detailed introduction of these metrics can be found in Appendix C.

To be more concrete, IRS evaluates the general interventional robustness of learned representations while UC and CG are extended from IRS. UC (Unconfoundedness) evaluates how well distinct generative factors are captured by distinct sets of latent dimensions with no overlap. CG (Counterfactual Generativeness) measures how robust a certain latent variable is against the others. IOSS measures whether learned random variables have independent supports as a necessary condition inferred from the definition of causal disentanglement. $D$ score measures disentanglement in the non-causal definition.

## 5.2 Experimental results

**(Q1) cdVAE significantly outperforms various baselines in end-to-end measurement.**
We compare our cdVAE with baseline models in the image generation task on the CelebA, and Candle datasets. In CelebA dataset, we have **C** set to be whether there are eyeglasses in the image, i.e., $|\mathbf{C}| = 2$. In the Candle dataset, we have $|\mathbf{C}| = 4$, with 2 objective hues and two wall hues combined together. More details can be found in Appendix C.

As shown in Table 1, cdVAE are evaluated by several groups of metrics. Reconstruction error measures how well the images are reconstructed in an end-to-end fashion while the others are measurements of how disentangled the latent factors are: D score is from the non-causal perspective while UC, UG and IOSS are from the causal perspective. For the celebA dataset, we do not measure the UC and CG score as it requires ground truth generative factors or access to the true generative process. Our method outperforms all baselines on these metrics except for the D score. Note the D score is only an effective measurement when there are no confounders in the observational dataset, as it requires each latent variable not to contain information about the others. Under the confounded dataset such as CelebA and Candle, the ground truth generative factors are correlated, resulting in a low D score with our model as expected.

**(Q2.1) cdVAE are more robust under distribution shifts.**
We conduct the task of shape classification on the 3dshape dataset with distribution shift and use the classification accuracy as a metric for out-of-distribution generalization[28]. Specifically, in the source distribution, we sample a certain percentage of images in which the object hue is correlated with the object shape (i.e., red objects are cubes). The rest of the images are evenly generated by

disentangled factors, while in the target domain, all images are generated by disentangled factors. The proportion of highly correlated data is denoted by *shift severity*. For example, shift severity = 0.4 means that 40% of training images are sampled under preset correlation between object hue and object shape.

We train cdVAE, $\beta$-VAE, and CAUSAL-REP using images from the source set, with decoders being replaced by classifiers. The trained classifier is then tested on the targets set. We report the classification accuracy on the target set and the performance drop in Figure 5. We could observe that cdVAE is more robust than other baselines under distribution shift as it has the lowest performance drop and the highest target set accuracy.

**An intuitive explanation of how C-Disentanglement improves the OOD generalization.** Suppose we have a fruit dataset and the goal is to classify fruits. In the training set, a large amount of apples are red and round. If assuming unconfoundedness and requires statistical independence, it is possible that a latent variable encodes whether this object is round-and-red. As a result, in the test set, if there is a green apple, it will fail the test and may not be able to be identified as an apple. However, under the framework of C-Disentanglement, color and shape are disentangled in the latent space, and the shape and color contribute to the classification separately, resulting in a higher probability for the green apple also to be identified as an apple.

Table 2: Compare how ground truth generative factors are recovered (MIC/TIC) and how disentangled they are in the latent space in classification on 3dshape dataset with shift severity = 0.5.

| Methods | MIC ↑ | TIC ↑ | IRS ↑ |
|---|---|---|---|
| VAE | 21.9 | 12.1 | 0.82 |
| $\beta$-VAE | 22.1 | 12.4 | 0.85 |
| FactorVAE | 24.3 | 15.6 | 0.89 |
| CAUSAL-REP | 26.8 | 16.1 | 0.88 |
| cVAE | 22.4 | 12.4 | 0.84 |
| GMVAE | 23.2 | 12.8 | 0.81 |
| cdVAE | **31.9** | **20.2** | **0.89** |

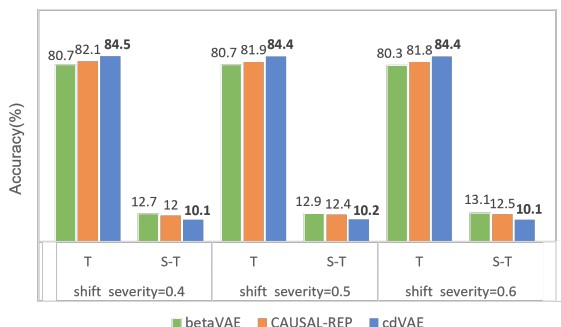

Figure 5: Compare cdVAE with $\beta$-Vae, CAUSAL-REP on classification under distribution shift. T represents accuracy on the target data, S-T represents the performance drop when the classifier trained on the source data is directly tested on the target data.

**(Q2.2) cdVAE better discovers the ground truth generative factors.**

In the task of shape classification, we further examine how well the learned representations approximate the ground truth generative labels and to what extent they are causally disentangled. Table 2 shows that cdVAE outperforms all baselines in approximating the generative factors and the disentanglement.

### 5.3 Ablations studies

We further conduct ablation studies to show that providing inductive bias indeed improves the discovery of generative factors in the latent space. Concretely, we compare cdVAE with conditional VAE (cVAE) [12] and GMVAE [7]. Compared with vanilla VAE, the method proposed uses label information to provide inductive bias to confounders for partitioning the observational dataset. cVAE has the label information compared with vanilla VAE and GMVAE is a VAE modelled by a mixture of Gaussian. As shown in Table 1, cdVAE has universally better results, showing the necessity of introducing bias to factors discovered in the latent space. We also investigate how choice of **C** affect the model performance and how to adapt C-Disentanglement to the existing method aiming for statistical independent, as shown in appendix C.

## 6 Related work

**Disentangled Representations** The pursuit for disentangled representation can be dated to the surge of representation learning and is always closely associated with the generative process in modern machine learning, following the intuition that each dimension should encode different features.

[6] attempts to control the underlying factors by maximizing the mutual information between the images and the latent representations. [8] propose a quantitative metric with the information theory. They evaluate the disentanglement, completeness, and informativeness by fitting linear models and measuring the deviation from the ideal mapping. [9, 11, 5, 18] encourage statistical independence by penalizing the Kullback-Leibler divergence (KL) term in the VAE objective. However, the non-causal definitions of disentanglement fail to consider the cases where correlated features in the observational dataset can be disentangled in the generative process. Such a challenge is well-approached through a line of research from the causal perspective.

**Causal Generative Process.** Causal methods are widely used for eliminating spurious features in various domains and improving understandable modelling behaviours[25, 26, 14]. It is not until [23] that it was introduced for a strict characterization of the generative process. [23] first provided a rigorous definition of a causal generative process and the definition of disentangled causal representation as the non-existence of causal relationships between two variables, i.e., the intervention on one variable does not alter the distribution of the others. The authors further introduce *interventional robustness* as an evaluation metric and show its advantage on multiple benchmarks. [21] follow the path of [23] and further propose two evaluation metrics and the Candle dataset. The confounded assumption allows for correlation in the latent space without tempering with the disentanglement in the data generative. Despite effective evaluation tools, there is still a missing piece on how to infer a set of causally disentangled features. Using the proposed evaluation metric as regulation, the model implicitly assumes unconfoundedness and it falls back to finding statistical independence in the latent space. The problem of unrealistic unconfoundedness assumption is identified by [24]. They assume that confounders exist but they are unobservable. They further propose an evaluation metric considering the existence of confounders, that causally disentangled latent variables have independent support measured by the IOSS score. Similar to the evaluation metrics introduced in [23, 21], IOSS is also a necessary condition of the causal disentanglement. More importantly, as in previous work focusing on obtaining statistical independence, such a regulation suffers from the identifiability issue.

**Weak Supervision for Inductive Bias.** The identifiability issue in unsupervised disentangled representation learning is first identified in [16]. Specifically, they show from the theory that such a learning task is impossible without inductive biases on both the models and the data. Naturally, a series of weak-supervised or semi-supervised methods [4, 1, 2] are proposed with a learning objective of statistical independence or alignment. In this paper, we take a step further for the confounding assumption, assuming that the confounders are observable with proper inductive bias so that the latent representation can be better identified. We, similarly, adopt partial labels of the dataset as the supervision signal. We treat the labels as a source of possible confounders and allow the learning of correlated but causally disentangled latent generative factors to be learned.

## 7 Conclusion and discussions

In this paper, we recognize the importance of bringing in confounders with proper inductive bias into the discovery of causally disentangled latent factors. Specifically, we propose a framework, C-Disentanglement, that introduces the inductive bias from the domain knowledge and a method, cdVAE, to identify the generative factors. Although cdVAE helps with discovering generative factors under confounders, the learning process relies on the domain knowledge, the choice of $C$, and that $C$ has to be discrete/categorical and have a finite number of realizations. How to generalize to continuous $C$ or select effective $C$ from the available knowledge set is beyond the scope of this paper and will be investigated in future studies. We hope this work could bring attention to the importance of inductive bias of confounders and inspire future works in this direction. In addition, better capturing of generative factors could bring a more controlled generative process and positive social impact.

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

# A  Introduction of do calculus.

Do-calculus consists of three rules that help with identifying causal effects.

**Rule A.1** (Insertion/deletion of observations)**.**

$$P(y|do(x), z, w) = P(y|do(x), w) \quad \text{if} \quad (Y \perp\!\!\!\perp Z|X, W)_{G_{\overline{X}}} \tag{13}$$

**Rule A.2** (Action/observation exchange)**.**

$$P(y|do(x), do(z), w) = P(y|do(x), z, w) \quad \text{if} \quad (Y \perp\!\!\!\perp Z|X, W)_{G_{\overline{X}\underline{Z}}} \tag{14}$$

**Rule A.3** (Insertion/deletion of actions)**.**

$$P(y|do(x), do(z), w) = P(y|do(x), w) \quad \text{if} \quad (Y \perp\!\!\!\perp Z|X, W)_{G_{\overline{XZ(W)}}} \tag{15}$$

where $G_{\overline{X}}$ is the graph with all incoming edges to $X$ being removed, $G_{\underline{W}}$ is the graph with all outcoming edges to $W$ being removed, and $Z(W)$ is the set of $Z$-nodes that are not ancestors of any $W$-node.

Intuitively, Rule A.1 states when an observant can be omitted in estimating the interventional distribution, Rule A.2 illustrates under what condition, the interventional distribution can be estimated using the observational dataset, and Rule A.3 decides when we can ignore an intervention.

# B  Proofs

## B.1  Proof of Proposition 2.1

**Proposition B.1.** *Let $Z_1$ and $Z_2$ be two random variables, $\mathbf{C}^*$ be the ground truth confounder set. If $\mathbf{C}$ is a superset of or is equivalent to $\mathbf{C}^*$, i.e., $\mathbf{C}^* \subseteq \mathbf{C}$, with $c$ being a realization of $\mathbf{C}$, we have*

$$P(Z_2|do(Z_1)) = \sum_{c \in \mathbf{C}} P(Z_2|Z_1, \mathbf{C} = c)P(\mathbf{C} = c) \tag{16}$$

*if no $C \in \mathbf{C}$ is a descendent of $\mathbf{Z}$.*

*Proof.*

$$P(Z_2|do(Z_1)) = P(Z_2|do(Z_1), \mathbf{C})P(\mathbf{C}|do(Z_1))$$

$$P(Z_2|do(Z_1), \mathbf{C}) \stackrel{\text{Rule A.2}}{=\!=\!=\!=\!=} P(Z_2|Z_1, \mathbf{C})$$

$$P(\mathbf{C}|do(Z_1)) \stackrel{\text{Rule A.3}}{=\!=\!=\!=\!=} P(\mathbf{C})$$

$$P(Z_2|do(Z_1)) = \sum_{c \in \mathbf{C}} P(Z_2|Z_1, \mathbf{C} = c)P(\mathbf{C} = c)$$

$\square$

## B.2  Proof of Theorem 4.1

**Theorem B.2.** *Suppose that the latent variable $\mathbf{Z}$ on dataset $\mathbf{X}$ given $\mathbf{C} = c$ is Gaussian $\mathcal{N}(\mu^c(\mathbf{X}), \Sigma^c(\mathbf{X}))$. Specifically,*

$$P(\mathbf{Z}|\mathbf{C} = c, \mathbf{X}) = (2\pi)^{-D/2} \det(\Sigma^c)^{-1/2} \exp\left(-\frac{1}{2}(\mathbf{Z} - \mu^c)^\mathsf{T}(\Sigma^c)^{-1}(\mathbf{Z} - \mu^c)\right),$$

*where $\mathbf{Z} \in \mathbb{R}^D$. If $\Sigma^c(\mathbf{X})$ is diagonal for all $c$, we have*

$$l_c = \sum_{i=1}^{D} d(\mathbb{E}(Z_i|do^c(Z_{-i}), \mathbf{X}), \quad \mathbb{E}(Z_i|\mathbf{X})) = 0. \tag{17}$$

*Proof.* We suppose that

$$P(\mathbf{Z}|\mathbf{C} = c, \mathbf{X}) = (2\pi)^{-D/2} \det(\Sigma^c)^{-1/2} \exp\left(-\frac{1}{2}(\mathbf{Z} - \mu^c)^{\mathsf{T}}(\Sigma^c)^{-1}(\mathbf{Z} - \mu^c)\right) \quad (18)$$

where we omit $\mathbf{X}$ for simplicity and $D$ is the dimension of $\mathbf{Z}$ for any given $c$. By definition of $l_c$ (Equation (9)) and proposition 2.1,

$$l_c = \sum_{i=1}^{D} d\left(\mathbb{E}[Z_i|do^c(Z_{-i}), \mathbf{X}] - \mathbb{E}[Z_i|\mathbf{X}]\right) \quad (19)$$

$$= \sum_{i}^{D} d(E[Z_i|Z_{-i}, \mathbf{X}, C = c], E[Z_i|\mathbf{X}, C = c]) \quad (20)$$

$$= \sum_{i}^{D} d(E[Z_i^c|Z_{-i}^c], E[Z_i^c]) \quad (21)$$

where we denote $\mathbf{Z}^c = [\mathbf{Z}|\mathbf{X}, C = c]$ for simplicity. Notice that $\mathbf{Z}^c \sim \mathcal{N}(\mu^c, \Sigma^c) \in \mathbb{R}^D$, we therefore know that the conditional distribution of any subset vector $Z_k^c$, given the complement vector $Z_j^c$, is also a multivariate Gaussian distribution [22]

$$Z_k^c|Z_j^c \sim \mathcal{N}(\mu_{k|j}^c, \Sigma_{k|j}^c) \quad (22)$$

where

$$\mu_{k|j}^c = \mu_k^c + \Sigma_{k,j}^c(\Sigma_{j,j}^c)^{-1}(Z_j^c - \mu_j^c), \quad \Sigma_{k|j}^c = \Sigma_{k,k}^c - \Sigma_{k,j}^c(\Sigma_{j,j}^c)^{-1}\Sigma_{j,k}^c, \quad (23)$$

given that $\Sigma_{j,j}^c$ is nonsingular.

Hence we know that the first expectation in Equation (21) becomes

$$E[Z_i^c|Z_{-i}^c] = \mu_i^c + \Sigma_{i,-i}^c(\Sigma_{-i,-i}^c)^{-1}(Z_{-i}^c - \mu_{-i}^c) \quad (24)$$

assuming that $\Sigma_{-i,-i}^c$ is nonsingular. Since $\mathbb{E}[Z_i^c] = \mu_i^c$, the loss $l_c$ can be written as

$$l_c = \sum_{i}^{D} d(\mu_i^c + \Sigma_{i,-i}^c(\Sigma_{-i,-i}^c)^{-1}(Z_{-i}^c - \mu_{-i}^c), \mu_i^c). \quad (25)$$

We assume further that $\Sigma^c$ is a diagonal matrix. Therefore $\Sigma_{-i,-i}^c = \mathbf{0}$ is a zero row vector. Then

$$l_c = \sum_{i}^{D} d(\mu_i^c, \mu_i^c) = 0 \quad (26)$$

$\square$

# C Experimental Details

## C.1 Experimental Details

The experiments are conducted on 4 NVIDIA GeForce RTX 2080Ti. In each experiment, we repeat 5 times with different seeds and report the averaged results. In all experiments, only partial information on the ground truth confounder is provided. Specifically, for example, the 3dshape dataset, we first make some predefined rules, such as " 70% cubes are red". Then we generate 700 red cubes and 300 cubes in other colors. The generation process naturally divides the dataset into different subgroups, and we can thus explicitly control how inductive bias is provided, i.e., the grouping. In the celebA dataset, since we do not have access to the ground truth generative factors, so we assume any label sets only contain partial information.

Table 5: **Compare cdVAE with $\beta$-Vae, CAUSAL-REP on classification under distribution shift. T represents accuracy on the target data, S represents the performance on the target domain when the classifier trained on the source data is directly tested on the target data.**

| Methods | shift = 0.4 | | shift = 0.5 | | severity = 0.6 | |
|---|---|---|---|---|---|---|
| | Acc-S | Acc-T | Acc-S | Acc-T | Acc-S | Acc-T |
| CAUSAL-REP | 94.1±0.04 | 82.1±0.08 | 94.3±0.03 | 81.9±0.07 | 94.3±0.02 | 81.8±0.11 |
| $\beta$-VAE | 93.4±0.07 | 80.7±0.12 | 93.6±0.05 | 80.7±0.03 | 93.4±0.04 | 80.3±0.09 |
| cdVAE | 94.6±0.02 | 84.5±0.05 | 94.6±0.04 | 84.4±0.05 | 94.5±0.03 | 84.4±0.04 |

## C.2 Ablation study

We investigate how the choice of $\mathbf{C}$ affect the model performance and how to adapt C-Disentanglement to the existing method aiming for statistical independence, as shown in appendix C.

Table 3: Performance under different $\mathbf{C}$ on shape classification on 3dshape dataset, shift severity=0.5.

| Choice of $\mathbf{C}$ | Acc - T ↑ | IRS ↑ |
|---|---|---|
| $\mathbf{C} = \emptyset$ | 79.2 | 0.82 |
| $\mathbf{C} = \mathbf{C}^*$ | 88.2 | 0.89 |
| partial $\mathbf{C}^*$ | 84.5 | 0.87 |

Table 4: Adapt cdVAE to existing methods. The IOSS and Reconstruction loss are measured based on image generation task and the performance drop is measured on shape classification on the target domain under shift severity=0.5.

| Methods | IOSS ↓ | Recon ↓ | Acc-T ↑ |
|---|---|---|---|
| IOSS | 0.14 | 0.12 | 81.8 |
| cdVAE + IOSS | 0.12 | 0.08 | 84.5 |

**C-Disentanglement improves the learning of ground truth generative factors under a reasonable choice of label set.** To understand how the choice of $\mathbf{C}$ affects the performance, we repeat the shape classification task with different choices of $\mathbf{C}$ under 50% shift severity. When with $\mathbf{C}$, we assume the generative process is unconfounded, and cdVAE degrades to the vanilla VAE model. With partial $\mathbf{C}$, we partition the data according to only 2 values of the shifting variables instead of 4. With full $\mathbf{C}$, we provide the full confounders. As shown in Table 3, even partial information of the confounders improves the model performance in OOD generalization and obtains more robust latent representations.

**Adapting C-Disentanglement to existing works further improve their performance.** We compare the performance between regulation through IOSS[24] and cdVAE + IOSS in image generation and classification tasks on 3dshape dataset. In the cdVAE + IOSS, we apply additional regularization terms based on the $\mathbf{Z}$. The results show that C-Disentanglement framework could further improve the performance with desired level of inductive bias given.

## C.3 Pseudo-code

## C.4 Additional Experimental Results

The classification accuracy on both the source and the target distribution with variance is given in the table below.

**Algorithm 1** Train a VAE such that the latent representation is causally disentangled

---

**Input:** Number of labels $N_C$, training data $\mathbf{X}$ with labels $c$, ratio of each categories/confounders $P(\mathbf{C} = c)$ in the training set, dimension of latent space $D$

1: **for** $x \in \mathbf{X}$ **do**
2:      **for** $c \in \mathbf{C}$ **do**
3:         Define $\mathbf{Z}^c = [\mathbf{Z}|x, C = c]$, and obtain from encoder $\mathbf{Z}^c \sim \mathcal{N}(\mu^c(x), \Sigma^c(x))$ for each $c$, assuming $\Sigma^c(x)$ to be diagonal matrix:

$$\Theta_{enc}^c(x) = [\mu^c, \texttt{diag}(\Sigma^c)] \in \mathbb{R}^{2d}, \quad \mu^c \in \mathbb{R}^d, \quad \texttt{diag}(\Sigma^c) \in \mathbb{R}^d \tag{27}$$

4:         Sample from $\mathbf{Z}^c \sim \mathcal{N}(\mu^c(x), \Sigma^c(x))$:

$$\mathbf{Z}^c = \mu^c + (\Sigma^c)^{\frac{1}{2}} \epsilon^c, \epsilon^c \sim \mathcal{N}(\mathbf{0}, I) \tag{28}$$

5:         Parametrize $\pi^c \sim \mathcal{N}(\mu_{\pi^c}(x), \sigma_{\pi^c}(x)) \in \mathbb{R}$ with neural network.
6:         Regulate the covariance matrix to be identity matrix with KL divergence

$$D_{KL}^c = \frac{1}{2} \left[ \log \frac{1}{\det \Sigma^c} - D + \texttt{tr}(\Sigma^c) \right] \tag{29}$$

7:      **end for**
8:      Normalize $\Pi_C = (\pi^{c_1}, ..., \pi^{c_{N_C}})$ such that $\|\Pi_C\|^2 = 1$.
9:      Compute classification loss between $\Pi_C$ with label $c$:

$$\mathcal{L}_{cls} = H(\Pi_C, c) \tag{30}$$

10:      Let $\mathbf{Z}(x) = \sum_{c \in \mathbf{C}} \pi^c \mathbf{Z}^c(x)$, and obtain the reconstructed sample from decoder: $x' = \Theta_{dec}(\mathbf{Z}(x))$. Compute reconstruction loss for $\mathbf{Z}(x)$:

$$\mathcal{L}_{rec} = \texttt{mse}(x', x) \tag{31}$$

11:      Compute total loss

$$\mathcal{L}_{total}(x) = \mathcal{L}_{rec} + L_{cls} + \sum_{c \in \mathbf{C}} D_{KL}^c \tag{32}$$

and update encoders and decoders.
12: **end for**

---

