# OpenReview forum: "C-Disentanglement: Discovering Causally-Independent Generative Factors under  an Inductive Bias of Confounder"
_NeurIPS.cc/2023/Conference — NeurIPS 2023 poster_

### Official Review · Reviewer_S7Fk · 2023-07-06

**Soundness:** 3 good
**Presentation:** 2 fair
**Contribution:** 2 fair
**Rating:** 5
**Confidence:** 3

**Summary:**

The paper proposed Confounded-Disentanglement a method that makes use of inductive bias of a confounder by leveraging labels/knowledge from domain expertise. The aim is to recover the true/causal generative factors for the observed data which can recover a more meaningful latent representation that can be further useful for downstream tasks especially in the case of distribution shifts.

**Strengths:**

- Originality: the authors propose a new way of leveraging observed confounders to learn disentangled representations.
- Quality: the manuscript includes both theoretical and empirical evidence for the claims.
- Clarity: I appreciate the schematics and toy running example to illustrate the intuition of the method.
- Significance: Better latent representations useful for overcoming domain generalisation challenges is timely and important contribution.

**Weaknesses:**

The problem of disentanglement, learning meaningful representations in general is an important , unresolved problem and I do agree that when additional information is available, we should make use of it, such as the scenario proposed by the authors of cdVAE. However, I do find the paper lacking in some recent relevant work and the overall presentation would benefit from including/positioning this method with regards to it. Please see more details in the question section.

**Questions:**

1. A recent stream of disentanglment that also leverages additional/auxiliary information, similar to ICA but even closer to cdVAE are the following [1, 2, 3]. In those cases, the additional information is treated as intervention, or environment. Could you please position your work wrt that? What makes a confounder different than their auxiliary variable?
2. What are the advantages of interpreting the additional variable as confounder and not intervention?
3. There are other metrics for measuring the quality of disentangled representations that haven't been used R2, MCC for example
4. Have you tried the controlled generative process under your framework? What could you conclude from the results?

[1] Lachapelle, Sébastien, et al. "Disentanglement via mechanism sparsity regularization: A new principle for nonlinear ICA." Conference on Causal Learning and Reasoning. PMLR, 2022.

[2] Khemakhem, Ilyes, et al. "Variational autoencoders and nonlinear ica: A unifying framework." International Conference on Artificial Intelligence and Statistics. PMLR, 2020.

[3] Lu, Chaochao, et al. "Invariant causal representation learning for out-of-distribution generalization." International Conference on Learning Representations. 2021.


---- Update after Author response ----
I thank the authors for the detailed and timely responses, they have mostly addressed my concerns, so I increase my score to BA.

**Limitations:**

Yes.

---

> ### Author Rebuttal · Authors · 2023-08-08
>
>
> We thank Reviewer S7Fk for the effort of reviewing our paper. We are encouraged to see that the reviewer appreciates the quality and the significance of our work.
>
>
> Most of the concerns and questions we believe are due to the confusions from various definitions of disentanglement in existing works. Below, our response clarifies the fundamental differences between our work and the works listed by the reviewer. Specifically, we emphasize our contribution of showing how to use confounders to tackle the mystery of learning semantically meaningful latent representations.
>
> Below we address the concerns of reviewer S7fk.
>
> ---
>
> > [Weakness and Q1] How do the methods [1,2,3], listed by reviewer, differ from our work?
>
> We thank the reviewer for the valuable suggestions of adding this line of works to the related work section and we will revise accordingly. However, we would like to point out that the works listed here [1, 3] are based on a nonlinear ICA method in [2]. They address different problems compared to our work.
>
> 1. The goal of [2] is to obtain conditional factorizable $p(z|u)$ while ours is to obtain causally disentangled $z$. **These two goals are irrelevant and are not mutually exclusive**.
>
>     - Let $z$ be the latent variable, $u$ be the auxiliary variable. The goal of non-linear ICA [2] is to learn $\tau$, $\lambda$ such that $$p_{\tau, \lambda} = \Pi_i\frac{Q_i(z_i)}{constant}exp[\sum_{j=i}^k \tau_{i,j}(z_i)\lambda_{i,j}(u)]\quad (1)$$
>     - If we further require $z$ to be a two dimensional vector, and $u$ only has one realization. Omitting the normalization constant in the first equation, Eq (1) becomes $$p(z_1, z_2 | u) = Q_1(z_1)\text{exp}(\tau_1(z_1)\lambda_1(u))Q_2(z_2)\text{exp}(\tau_2(z_2)\lambda_2(u))\quad (2)$$
>     - Our goal is to learn causally disentangled $z$ which satisfies
>
>     $$p(z_i | do(z_{-i})) = \sum_{u}p(z_i | z_{-i}, u) = p(z_i) \quad \forall i\quad (3)$$
>
>     - under the same assumption of $u$ and $z$, causally disentangled $z$ in our paper satisfies $$p(z_1 | z_2, u) = p(z_1) \text{ and } p(z_2 | z_1, u) = p(z_2)\quad (4)$$
>
>
>     **It can be easily observed that, the causal disentanglement property in Eq (4) above is irrelevant to the goal of ICA-based methods (2). In fact, it has also been pointed out in [1] that methods in [2] cannot achieve causal disentanglement.**
>
> 2. Using [2] as a backbone, [1] and [3] introduce causal representation learning methods. However, our method is different from theirs due to the following reasons:
>     - Although [1] also aims for causal disentanglement, it assumes the dataset is **unconfounded** and specializes in dealing with sparsity regulation of sequential data.
>     - [3] does not aim for causal disentanglement representations; instead it finds an invariant causal graph across different domains.
>
>
>
> ---
>
> > [Q2] What are the advantages of interpreting the additional variable $u$ as confounder and not intervention?
>
>
> - As illustrated above, confounder $u$ in Eq (3,4) is conceptually different from the auxiliary variable $u$ in Eq (1,2) (in [1,2,3] pointed out by the reviewer). Auxiliary variables are weak supervision labels conditioned on which we get factorial $p(z|u)$, whereas confounder is a concept in causal inference which is the common parents of generative factors, to allow causally independent factors be statistical dependent in the latent space.
>
>
> - However, $u$ can be interpreted as an inductive bias for both the confounder and the intervention (mathematically, $u$ and $z$ satisfy both (1) and (3)). We defer it to future work to investigate the benefit of combing these two.
>
> - Below, we explain the benefit of modeling the confounder.
>     - Confounder allows causally independent but correlated generative factors to be detected. Assuming unconfoundedness, on the contrary, requires global statistical independence among variables.
>     - For instance, a fruit dataset is generated from three generative factors (color,shape,size). The confounder is the fruit type. Although color, shape and size are causally independent, due to the type of the fruit, they are correlated in the observational dataset (banana is yellow, apple is red and green). But if you fix the fruit type, we could observe that color and shape are not correlated.
>
>
> ---
>
> > [Q3] There are other metrics for measuring the quality of disentangled representations that haven't been used R2, MCC for example.
>
> We choose evaluation metrics to measure
> - (1) whether we can recover the ground truth generative factors in the synthetic datasets where ground truth factors are available, as the metrics 'MIT' and 'TIC' used in Table 1.
> - or (2) whether the recovered factors are causally disentangled (how changing one factor affect the others) in real-world datasets where the ground truth factors are not available, as the metrics 'IRS', 'CG'/'UC' and 'IOSS' used in Table 1 and 2.
>
> MCC and R2 do not fall into these categories and do not measure the quality of causal disentanglement.
>
>
> ---
>
> > [Q4] Apply our method to controlled generative process?
>
> We have used the CG scores [21] (the higher, the better) to measure the controlled generation of the counterfactual images. The CG metric evaluates how well the learned generative factor set is able to generate counterfactual instances in a flexible and controlled manner. Our method achieved higher CG scores as shown in Table 1.
>
>
> ---
>
> We thank the reviewer again for the time and effort. We have addressed all the concerns raised by the reviewer; most of them are due to different definitions of disentanglement in existing literature. We hope Reviewer S7Fk could consider revising the scores accordingly. Please do not hesitate to reach back if the reviewer has any questions.

---

> > ### Author Response · Authors · 2023-08-18
> > **Additional questions**
> >
> > Thanks again for your time and your insightful comments.  Is there any remaining concern about our paper? We are more than delighted to address any questions you may have.

---

> ### Comment · Area_Chair_7g8n · 2023-08-19
> **discuss with authors**
>
> Dear reviewer S7Fk,
>
> The author reviewer discussion deadline is approaching. Could you please read the rebuttal to see if you need further clarifications?
>
> Thanks, AC

---

> ### Author Response · Authors · 2023-08-20
> **Awaiting your feedback to our rebuttal**
>
> Dear Reviewer S7Fk,
>
> We appreciate your insightful feedback and suggestions. We have provided an in-depth comparison between our works and the works listed. We have shown that the confusion comes from various definitions of disentanglement, and the problem-setting and goals are fundamentally different. We have also provided additional evaluation according to your comments.
>
> We genuinely hope our explanations have clarified any ambiguities that you may have. If you feel our responses have adequately addressed your concerns, we'd be grateful for your feedback or any follow-up questions.  Since you also agreed that our work is solid, novel, and solves a significant problem, we would kindly ask you to consider re-evaluating the score.
>
> Best regards,
> Authors.

---

### Official Review · Reviewer_j8n4 · 2023-07-06

**Soundness:** 3 good
**Presentation:** 1 poor
**Contribution:** 2 fair
**Rating:** 5
**Confidence:** 3

**Summary:**

This paper is motivated by the assumption that a few semantically significant generative factors generate real-world data. Then they study causal independence in generative methods by introducing a new framework, C-Disentanglement, that discusses the identifiability issue of generative factors regarding the inductive bias of confounder to handle the disadvantage brought by the statistical independence assumption of previous methods. At last, the authors implement their method, cdVAE, and evaluate it on three datasets. The results show that cdVAE performs better in the latent space compared to the existing methods and performs better in downstream tasks.

**Strengths:**

The motivation is obvious, which emphasizes the importance of causal independence in generative methods and illustrates the advantages of causal independence for statistical independence, which is very interesting and important. To the best of our knowledge, the authors are the first to study how to explore causal independence confounders in generative methods.
This article has complete content. It conducts a thorough investigation of the background. Then it proposes a new framework and elaborates in detail. Moreover, it has several experiments on three datasets to demonstrate its advantages.

**Weaknesses:**

This article includes but is not limited to the following shortcomings:
1. There was much ambiguity about collaboration, which led me to rely a lot on my guesswork in understanding the article, like:

a) D in line 141 and D in equation 7 do not seem like the same thing but use the same symbol.

b) D_KL in Figure 4 and D in Equation 7 seem to represent KL divergence but in different symbols.

c) What is the meaning of \Pi? I do not find its definition. Is that represent the soft assignment of samples? If it is, what is the connection between \Pi and \pi?

2. In Table 1, you evaluate your method on CelebA and Candle datasets. However, the measurement metrics between these datasets are different. You claim that the CelebA dataset does not have ground truth factors. However, the experiments on only two datasets cannot fully illustrate your methods' advantages, especially since CelebA does not have ground truth factors.
3. There are many kinds of generative methods, like the diffusion model. There is no comparison between your method and other generative methods.
4. The baseline vanilla VAE (2013), \beta-VAE (2016), FactorVAE (2018), CAUSAL-REP (2021), which do not compare, cVAE (2013), GMVAE (2016), are out-of-date.

**Questions:**

1. I think implementing your ideas in VAE is a limitation for you. I hope you can implement your ideas in more generative methods.
2. The Related Work in the main paper is very similar to the Related Work in the appendix, so I doubt the significance of the Related Work in the appendix. However, I do not think this is a weakness of your paper, so I put this into Question.

**Limitations:**

The authors point out the limitation of their method at the end of the paper; that is, the choice of C is difficult but essential, which is the same as our understanding of their method.

---

> ### Author Rebuttal · Authors · 2023-08-08
>
> We want to thank Reviewer j8n4 for the effort of reviewing our paper and the recognition of the novelty and significance of the proposed method. We hope that our response and additional results could clarify doubts toward our work.
>
> ---
>
> > [Q1] Notation issue
>
> We would like to thank reviewer j8n4 for pointing out some notation issues in the narrative. We have carefully revised our manuscript accordingly to ensure the consistency of these notations.
>
> For questions listed here. $D$ in line 141 denotes the dimension of the latent representation and the $D$ in equation 7 should be changed to $D_{KL}$ to represent the KL divergence. For $\pi_c$ in equation 11, it is the  mixing coefficient as stated in line 220. $\pi_c = P(C = c|X)$ reads the probability of occurrence of $C = c$. And in line 223, $\pi_c \sim \mathcal{N}(\mu^{\pi_c}(X), \Sigma^{\pi_c}(X))$ where the $\mathcal{N}(\mu^{\pi_c}(X), \Sigma^{\pi_c}(X))$ should be $\mathcal{N}(\mu^{c}(X), \Sigma^{c}(X))$.
>
> ---
>
> >[Q2] Lack of ground truth for evaluation of causality.
>
> - In addition to CelebA real-world dataset, we also evaluate on two synthetic datasets, Candle and 3dshape, that have ground truth generative factors. The results (in Table 2 & Figure 5)  show that our method has better performance in obtaining causal disentanglement, can recover ground truth generative factors better, and is more robust to distribution shift.
>
> - CelebA, Candle & 3dShape datasets are commonly used benchmarks for evaluating causal representation learning methods. Our work is consistent with this line of works. For example, [23,24] tested their method on CelebA and their own small-scale synthetic dataset. [21] evaluated their method on Candle. [R1] also tested on one real-world dataset and one synthetic dataset.
>
> [R1] Lachapelle, Sébastien, et al. "Disentanglement via mechanism sparsity regularization: A new principle for nonlinear ICA." Conference on Causal Learning and Reasoning. PMLR, 2022.
>
> ---
>
> > [Q3] Comparison with other generative models like Diffusion.
>
> - Our contribution is not to introduce a generative model, but to introduce a general representation learning framework, C-Disentanglement, that can be applied to all models that have a latent space.
>
> - Our representation learning framework aims for capturing causally disentangled representations in the latent space. The contribution is significant especially when the ground truth generative factors are *statistically correlated* but *causally disentangled*, which cannot be captured by exiting methods (since the existing methods learn *statistically independent* latent features).
>
> **Additional baselines: latent diffusion[R2] and InfoGAN[R3]**
>
> As suggested by the reviewer, we have further added two additional baselines.
>
> - [Additional generative model, latent diffusion] We have added the latest latent diffusion model [R2] as a baseline.
> - [Additional generative model, IB-GAN] We also have added another baslie: IB-GAN [R3]. It is the latest GAN-based method on disentangling latent representation although the definition of disentanglement in this paper differs from ours.
> - **The result is given in Table 3 of the 1-page pdf.**
>
> [R2] Rombach, Robin, et al. "High-resolution image synthesis with latent diffusion models." Proceedings of the IEEE/CVF conference on computer vision and pattern recognition. 2022.
>
> [R3] Jeon, Insu, et al. "Ib-gan: Disentangled representation learning with information bottleneck generative adversarial networks." Proceedings of the AAAI Conference on Artificial Intelligence. Vol. 35. No. 9. 2021
>
> ---
>
> > [Q4] Baseline model out-of-date?
>
> Since our contribution lies in learning causally disentangled representation and most of the baselines use VAE as the backbone, we also choose VAE for a fair comparison. We believe the baseline models are not out-of-data, since they are SOTA causal disentangled representation learning methods.
> - For example, We compare our method with CAUSAL-REP [23], which is the sota method on obtaining causal disentanglement.
> - We compare with cVAE and GMVAE as ablation studies, because cVAE also has additional label information, GMVAE is also a mixture of Gaussian model.
> - By showing that ensuring C-disentangled representation under confounder modeling improves VAE, we anticipate similar improvement in other models.
>
> ---
> > [Q5] Implementing the ideas only in VAE is a limitation?  Implement your ideas in more generative methods?
>
> We thank Reviewer j8n4 for the recognition of the significance and soundness of our proposed idea. We would like to clarify that, the framework we proposed in this paper, Confounded Disentanglement (C-Disentanglement) is not confined in the VAE but a general framework that can be used in common generative models and even outside the scope of generative models.
>
> Our cdVAE, under this premise, is an algorithm and a practical solution that utilizes this framework. We choose VAE because  (1) it is one of the mainstream formulation of the generation problem, and (2) with the Gaussian assumption, perfect C-Disentanglement can be theoretically obtained in an easy-to-implement and effective way as stated in Theorem 4.1 (line 213).
>
>
> Applying the C-Disentanglemt in generative models is a non-trivial task and does not have a common algorithmic solution. For example, diffusion models do not have a latent space. The development of corresponding algorithms under different generative models may go beyond the scope of this paper and we'd like to save it to future works.
>
> ---
>
> > [Q6] [Related work in the main paper and appendix]
>
> We have shortened the related work section in the main paper due to the page limit, and put a complete version to the appendix.
>
> ---
> We thank the reviewer again for the time and effort. We have addressed all the concerns raised by the reviewer. We hope Reviewer j8n4 could consider revising the scores accordingly. Should Reviewer j8n4 have any other questions or concerns, please do not hesitate to reach back.

---

> > ### Author Response · Authors · 2023-08-18
> > **Additional questions?**
> >
> > Thanks again for your time and your comments.  Is there any remaining concern about our paper? We are more than delighted to address any questions you may have.

---

> ### Comment · Area_Chair_7g8n · 2023-08-19
> **discuss with authors**
>
> Dear reviewer j8n4,
>
> The author reviewer discussion deadline is approaching. Could you please read the rebuttal to see if you need further clarifications?
>
> Thanks, AC

---

> ### Author Response · Authors · 2023-08-20
> **Awaiting your feedback on our rebuttal.**
>
> Dear Reviewer j8n4,
>
> Thank you for your time and effort in reviewing our paper and providing valuable comments. We have provided additional baselines according to your comments and have discussed in the rebuttal the potential confusion and concerns that you may have. We genuinely hope our explanations have clarified any ambiguities that you may have. If you feel our responses have adequately addressed your concerns, we'd be grateful for your feedback or any follow-up questions,  and would kindly ask you to consider re-evaluating the score.
>
> Best regards,
> Authors.

---

> ### Comment · Reviewer_j8n4 · 2023-08-21
>
> Thanks for the rebuttal, most of my concerns are addressed. The presentation can be much improved. I will change my rating to BA.

---

### Official Review · Reviewer_TdLu · 2023-07-07

**Soundness:** 2 fair
**Presentation:** 3 good
**Contribution:** 3 good
**Rating:** 7
**Confidence:** 4

**Summary:**

This paper delves into the domain of representation learning, focusing on the discovery of meaningful generative factors within a latent space. The authors propose the C-Disentanglement framework, which incorporates the inductive bias of confounders through labels or knowledge obtained from domain expertise. This framework enables the identification of generative factors even in the presence of confounding variables. The paper offers a comprehensive theoretical analysis and a practical approach to effectively identify these factors. Empirical results obtained from diverse settings validate the efficacy of the proposed method.

**Strengths:**

	This paper is very well written and easy to follow.
	The research addresses an important and fundamental question: how to identify Causally-Independent generative factors in the presence of common causes.
	The idea proposed in this work is simple but certainly make sense. And the sufficient experiments validate the effectiveness of the proposed method.


**Weaknesses:**

	The assumptions made in this paper are quite strict. One of the main concerns is that the paper assumes the probability distributions of the generative factors to be Gaussian distributions. This assumption may limit the applicability and generalizability of the proposed method to real-world scenarios where the underlying distributions may not strictly follow Gaussian patterns. Another strong assumption in this paper is the linear mixing of the generative factors. While this assumption might simplify the modeling process, it may not accurately capture the complex relationships and interactions between the generative factors in real-world scenarios. It is important for the authors to discuss the potential impact of this assumption on the performance and generalizability of the proposed method.
	The proposed inductive bias in this paper is hard to obtain. The utilization of domain expertise knowledge as an inductive bias is an expensive and non-scalable approach.
	The evaluations of the proposed method primarily focus on quantitative aspects, and incorporating qualitative evaluations would provide a more intuitive understanding of the method's performance.


**Questions:**

	How can we ensure that the number of C is adequate?
	In Line 215, Equation (10), the right-hand side of the equation involves several terms that are not enclosed within absolute value symbols. Does these terms are non-negative?


**Limitations:**

The authors adequately addressed the limitations of the work.

---

> ### Author Rebuttal · Authors · 2023-08-08
>
>
> We greatly appreciate Reviewer TdLu for the valuable feedback and suggestions. We are encouraged that Reviewer TdLu finds the problem setting is fundamental and important, and our paper in good clarity and quality. We address the concerns and discuss the insightful suggestions raised by the reviewer below.
>
> ---
>
> > [Weakness 1] The assumptions made in this paper are quite strict. One of the main concerns is that the paper assumes the probability distributions of the generative factors to be Gaussian distributions. This assumption may limit the applicability and generalizability of the proposed method to real-world scenarios where the underlying distributions may not strictly follow Gaussian patterns. Another strong assumption in this paper is the linear mixing of the generative factors. While this assumption might simplify the modeling process, it may not accurately capture the complex relationships and interactions between the generative factors in real-world scenarios. It is important for the authors to discuss the potential impact of this assumption on the performance and generalizability of the proposed method.
>
> We thank the reviewer for this insightful question and we will add a discussion to our paper for these assumptions.
>
> We would like to clarify that the framework we proposed in this paper, Confounded Disentanglement (C-Disentanglement) is not confined in the VAE but a general framework. It does not have assumptions on distributions of the observational dataset.
>
>
> **Gaussian prior**
>
>
> Our cdVAE, under this premise, is an algorithm and a practical solution that utilizes this framework. In cdVAE, we also do not assume the distribution of the observed data. In VAE-based methods, the latent is Gaussian, but there always exist a map from Gaussian to the observed data distribution that can be learned.
>
>
> **Linear mixing of the gaussian distributions**
>
>
> We want to clarify that the linearity does not come from our assumption but statistical derivation. The conditional probability $P(Z|X)$ can be factored into $$P(Z|X) = \sum_{c \in C} P(Z|X, \mathbf{C} = c)P(\mathbf{C} = c|X).$$
>
> Given our assumption that the latent variables are Gaussian for each labels, this reads
> $$ P(Z|X) = \sum_{c \in C} P(\mathbf{C} = c|X)\mathcal{N}(\mu^c(X), \Sigma^c(X)) $$ where $P(\mathbf{C} = c|X)$ is the mixing coefficient. To enable more flexibility, we model this coefficient as a Gaussian variable with its own mean and variance instead of using a hard assignment.
>
> ---
>
> > [Weakness 2] The proposed inductive bias in this paper is hard to obtain. The utilization of domain expertise knowledge as an inductive bias is an expensive and non-scalable approach.
>
> - We thank Reviewer for the raising the question of accessibility of the inductive bias. Empirically, while the ground truth $C^*$ is hard to obtain, it is not that hard to obtain some additional labels of a dataset. we observed improvements even with $C$ contains partial information about the ground truth confounder (as shown in appendix D.2).
>
> We have added more results on celebA with randomly-chosen labels as shown **in Table 2 in the 1-page pdf** for rebuttal. The results universally outperform the baseline models in both accuracy and casual disentanglement metrics.
>
> Although a challenging task, one way to automatically generate proper $C$ is to train a model for proper $C$ in an end-to-end fashion, or infer $C$ with the help of large pretrained models. We will save it to future works.
>
>
> ---
>
> > [Weakness 3] The evaluations of the proposed method primarily focus on quantitative aspects, and incorporating qualitative evaluations would provide a more intuitive understanding of the method's performance.
>
> We appreciate the reviewer for suggesting qualitative evaluations for more intuitive understanding of the methods. We currently have an ablation study on how different choices of $C$ (partial information, full information, no information about the ground truth $C$) affect the learning in appendix D.2 Table. We will add more qualitative evaluations in revised version.
>
> ---
>
>
> >[Q1]How can we ensure that the number of C is adequate?
>
> Thanks for this insightful question.
>
> - In our paper, we propose cdVAE under the framework of C-Disentanglement to learn causally independent representational variables in the latent space. Instead of enforcing global statistical independence among variables on the observational dataset, we partition the dataset into subsets according to realizations of $C$, and regulate these variables within each subset.
>
> - The number of $C$ determines how dataset are partitioned, thus an adequate $C$ is expected to leave enough samples in each subset for model inference and we now rely on domain expertise for the selection. Since one of the main messages of our paper is to encourage learning of causally disentanglement through modeling confounders, we save the automated selection of $C$ to future works.
>
> ---
>
>
> >[Q2] In Line 215, Equation (10), the right-hand side of the equation involves several terms that are not enclosed within absolute value symbols. Does these terms are non-negative?
>
> We apologize for the typo in equation 10.
>
> It should be $l_c=\sum_{i=1}^{D}d\left[\mathbb{E}(Z_i | do^c({Z_{-i}}), X),  \mathbb{E}(Z_i | X)\right] = 0$, where $d$ is the euclidean distance and it is non-negative.

---

> > ### Author Response · Authors · 2023-08-18
> > **Additional questions?**
> >
> > Thanks again for your time and your comments.  Is there any remaining concern about our paper? We are more than delighted to address any questions you may have.

---

> > > ### Comment · Reviewer_TdLu · 2023-08-20
> > >
> > > Thank the authors for the clarifications. I will keep my score.

---

> > > > ### Author Response · Authors · 2023-08-20
> > > >
> > > > Thank you again for your in-depth feedback and comments.

---

> ### Comment · Area_Chair_7g8n · 2023-08-19
> **discuss with authors**
>
> Dear reviewer TdLu,
>
> The author reviewer discussion deadline is approaching. Could you please read the rebuttal to see if you need further clarifications?
>
> Thanks, AC

---

### Official Review · Reviewer_Mytf · 2023-07-11

**Soundness:** 3 good
**Presentation:** 2 fair
**Contribution:** 3 good
**Rating:** 7
**Confidence:** 4

**Summary:**

The paper proposes a new framework called Confounded-Disentanglement (C-Disentanglement) to learn disentangled representations by using inductive bias of confounding. Instead of the common statistically independent assumption for latent distribution which is usually not true in real-world datasets, the proposed framework partitions the observed data distribution based on a label set C and regulates the latent distribution separately for these subsets.

The proposed framework is then implemented in a Variational AutoEncoder (VAE) setting, termed cdVAE. Practically, the proposed method models the latent distribution as a mixture of Gaussian model in which each individual Gaussian corresponds to a specific value of C.

The paper shows the effectiveness of the proposed method on the image generation task on CelebA and Candle datasets, and on a classification task under distribution shift on 3DShapes dataset.

**Strengths:**

- The paper extends research in disentangled representation learning by leveraging inductive bias of confounding that makes it more applicable to real-world datasets that contain correlations in generative factors.
- The proposed framework and method are technically sound.
- The results seem promising.
- The proposed method is evaluated on multiple tasks and multiple datasets which shows the robustness of the proposed method.

**Weaknesses:**

- The proposed method requires domain expertise knowledge to define the confounders. This can be easy in some cases but could be costly (in terms of time and cost) and error-prone in many cases. What would happen if an error is introduced in defining the label set? Also, how to distinguish from the cases where each subset of data (defined by a specific value of the label set) still contains correlations among generative factors e.g., in the example of Figure 3, if the imported apples contain large green apples and small yellow apples? Is it a requirement that the label set needs to be granular enough so that no correlations remain in the subset corresponding to each specific value of the label set?

- What cases from section 3 do the experiments from the paper fall in? Please discuss this in detail.

- The proposed method cdVAE effectively boils down to modeling the latent distribution as a mixture of Gaussians. How is it different from GMVAE [7] then?

- Could the authors explain in more detail how the objective in CausalVAE [24] is fundamentally different from this paper?

- The paper mentions that using a Mutual Information (MI) objective for latents is not sufficient/appropriate. Then, shouldn’t the evaluation metrics based on MI be avoided as well? Also, why not use DCI/D score in Table 2?

- Can the authors explain in a bit of detail how all the evaluation metrics for disentanglement are being calculated and why UC and CG scores can't be calculated for CelebA? Can't the attribute annotations in CelebA be used as a proxy for generative factors?

- The shape classification generalization experiment setting is a bit unclear. Can the authors please explain it again, maybe with an example?

- All the results need error bars.

**Questions:**

I have included all my questions that I would like the authors to answer in the weaknesses section above. The answers to all these issues are crucial to my final recommendation.

**Limitations:**

The authors have adequately addressed the limitations of the proposed method.

---

> ### Author Rebuttal · Authors · 2023-08-08
>
> We greatly appreciate Reviewer Mytf for the insightful feedback and suggestions. We are encouraged that Reviewer Mytf finds our work solid and promising. We address the concerns and questions raised by the reviewer below. Should Reviewer Mytf have any other questions or concerns, please do not hesitate to reach back.
>
> ---
>
> > Q1 Whether the label set needs to be precise and exact when providing the inductive bias.
>
> We thank the reviewer for this insightful and interesting question.
>
> - The proposed method is robust to errors, as using irrelevant information as confounder will not affect the learning. (Proposition 2.1 and Figure 3d).
> - When the confounder is not inclusive/granular enough, the generating factors can be partially recovered. In the reviewer's example, the shape can be disentangled from size and color.
> - We use partial $C$ in all our experiments. The results show that mild information about $C^*$ could help obtain robust and causally disentangled latent factors.
> - We show performance gain even with randomly selected $C$ CelebA in **Table 2 on the 1-page pdf** .
>
> ---
>
> > Q2 What cases from section 3 do the experiments from the paper fall in? Please discuss this in detail.
>
> - Except for our method, the rest of the methods assume unconfoundedness, which falls into **case 1** where $C=\emptyset$.
> - Our method in table 1 and table 2 contains partial information on the ground truth confounder **(case 2)**, as it is the most practical situation when applied to real-world applications.
> - We further conduct the ablation study to explore how different levels of inductive **(case1, 2, 3)** affect the final performance. The result is given in appendix D.2 table 3.
>
> ---
>
> > Q3 Difference between our work and GMVAE[7].
>
> The main difference lies in whether confounder is modeled.
>
> - In our method, we partition the dataset according to realizations of the confounder, and within each subset, we model the latent distribution as Gaussian and it obtains C-Disentanglement.
> - In GMVAE, the number of Gaussian is a hyper-parameter and there is no specific partition for each Gaussian distribution. Thus the causal disentanglement can not be guaranteed.
>
> ---
>
> > Q4 Difference between our work and CausalVAE[24].
>
> -  The goal of our work is to infer generative factors in the latent space, whereas the goal of CausalVAE is to use latent variables to encode any given labels.
> -  We require the latent variables to be causally disentangled, whereas CausalVAE tries to learn a causal graph of the given labels and requires the latent variables to follow the same graph structure.
> -  We explicitly model the confounder whereas CausalVAE assumed unconfoundedness.
>
>
> ---
>
> > Q5 The paper mentions that using a Mutual Information (MI) objective for latents is not sufficient/appropriate. Then, shouldn’t the evaluation metrics based on MI be avoided as well? Also, why not use DCI/D score in Table 2?
>
>
> We totally agree that the MI evaluation metrics should be avoided in measuring how well the latent variables being causally disentangled.
>
> - However, in table 2, we want to show how much information of the ground truth generative factors is captured by our learned latent representation as a whole. Therefore, we compute MIT and TIC. High mutual information among them indicates the learned set of latent variables better captures the ground truth generative factors.
> - We include $D$ in Table 1 to demonstrate that it cannot accurately reflect level of causal disentanglement of a representation.
> - We do not include $D$ as it is used to compare among latent variables, but not between the learned representation and the ground truth.
>
>
> ---
>
> > Q6 why UC and CG scores can't be calculated for CelebA with annotation attributes as a proxy of ground truth generative factors?
>
> We thank the reviewer for this question and we will also add the detailed explanation and calculation of all evaluation metrics to the appendix. Here we focus more on UC and CG score due to the word limit.
>
> - The UC metric evaluates how well distinct generative factors are captured by distinct sets of latent dimensions with no overlap. $UC = 1- \mathbb{E}[\frac{1}{S}\sum_{I,J}\frac{Z_I \cap Z_J}{Z_I \cup Z_J}]$ where $S$ is the number of pairs of ground truth generative factors $(G_I, G_J)$, $Z_I, Z_J$ is latent variables. With annotation attributes as a proxy of generative factors, we could calculate UC score. **The results are given in the 1-page pdf as Table 1**.
>
> - The CG metric evaluates how well the learned generative factor set is able to generate counterfactual instances in a flexible and controlled manner. $CG = \mathbb{E}[|ACE_{Z_I} - ACE_{Z_{-I}}|]$ where ACE is the averaged causal effect. With celebA dataset we are not able to generate any new images but only sample from it. Consequently, we are not able to obtain enough data points for counterfactual images, and thus could not calculate CG scores.
>
> ---
>
> > Q7 The shape classification generalization experiment setting is a bit unclear. Can the authors explain it again, maybe with an example?
>
> In shape classification experiment, we test how our cdVAE works under the distribution shift. In the source set, we set a confounder to bring in correlation and sample a certain amount of data accordingly. For example, the ground truth generative factors are size, shape and color, and the confounder is the type of objects, which requires a certain percentage cubes are red, cylinders are yellow and etc. This way, spurious correlation exists in the source set between the object shape and color. In the target set, images are generated with random combination of the generative factors (the will also be green cubes, or blue cubes which may not exist in the source set, and the distribution also changes). We train the model on the source and test it on both sets.
>
> ---
> > Q8 Error bar
>
> The error bar for Figure 5 is given in the appendix and We have added the error bars for table 1 and table 2 to the revised paper.

---

> > ### Author Response · Authors · 2023-08-18
> > **Additional questions?**
> >
> > Thanks again for your time and your comments.  Is there any remaining concern about our paper? We are more than delighted to address any questions you may have.

---

> ### Comment · Area_Chair_7g8n · 2023-08-19
> **discuss with authors**
>
> Dear reviewer Mytf,
>
> The author reviewer discussion deadline is approaching. Could you please read the rebuttal to see if you need further clarifications?
>
> Thanks, AC

---

### Author Rebuttal · Authors · 2023-08-08



### Summary of Rebuttal Discussion


We thank all reviewers for the valuable feedback and insightful questions. We are particularly encouraged that they consider our proposed framework and method fundamental, important and novel.

---

### Review Highlights

Reviewer Mytf finds our work "**technically sound**" and "**promising**".

Reviewer TdLu believes that "The research **addresses an important and fundamental question**" and "the sufficient experiments validate the effectiveness of the proposed method"

Reviewer j8n4 agrees that our work is **well motivated and novel**, in that "The authors are the first to study how to explore causal independence confounders in generative methods"

Reviewer S7Fk appreciates the originality, quality, clarity and the significance of our work, and mentions that our paper is a "**timely and important contribution**".

---

### Summary of Core Contributions

We would like to highlight our core contribution as follows:

1. We **recognize the identifiablility issue** of discovering generative factors in the latent space. That correlated but causally independent generative factors cannot be recovered by existing methods assuming unconfoundedness in modeling.
2. We accordingly introduce **a novel and general framework**, **C-Disentanglement**. It is the first framework that discusses how inductive bias of confounder could be explicitly provided via labels/knowledge from domain expertise.
3. We propose **an algorithm, cdVAE**, that discovers causally disentangled generative factors in the latent space. The algorithm sheds light on the easy injection of inductive bias into existing methods.

We hope our work could

- brings out new perspectives on discovering semantically meaningful latent representations,
- inspires theoretical and empirical ideas in modeling confounder and causal representation learning.


---

### Summary of Changes

We address individual questions of reviewers in separate responses.

Here we briefly outline the updates to the revised version of our paper based on the reviews.


**[Problem Formulation]**  We added a discussion of assumptions and the potential influence. (Reviewer TdLu)

**[Experiment]** We added several experimental results as follows

- baselines of Latent Diffusion and IB-GAN (Reviewer j8n4),
- UC scores on CelebA using annotation attributes as a proxy of the ground truth generative factors, and error bars in appendix (Reviewer Mytf)
- How randomly selected C affects the model performance on real-world celebA dataset. (Reviewer TdLu, Reviewer Mytf)

**[Related Work]** We added comparison between our work and another line of work with various definitions of disentanglement (Reviewer S7Fk) and remove similar parts in the appendix (Reviewer j8n4)

**[General]** We revised the notation for consistency and corrected typos. (Reviewer j8n4)

---

### Author Response · Authors · 2023-08-15

Thank you all for the constructive suggestions and the insightful questions. We have provided clarifications to all the questions and concerns raised in the reviews, and have conducted additional experiments according to the feedback. We were wondering if there are any remaining concerns and questions about our paper, and we are more than happy to address them.

---

### Decision · Program_Chairs · 2023-09-21

**Decision:**

Accept (poster)

**Comment:**

The paper introduces a novel framework called "Confounded-Disentanglement" (C-Disentanglement) for acquiring disentangled representations by considering possible confounding in the latent space. To solve this ill-posed problem, this papers introduces proper inductive bias to identify the causally-disentangled factors. The paper provides empirical evidence of the effectiveness of this approach in tasks such as image generation using CelebA and Candle datasets, as well as in addressing a classification task involving distribution shift using the 3DShapes dataset.

This paper tackes the disentanglement problem in a more practical setting and the proposed method is technically sound. The paper's contribution is expected to be highly beneficial and influential in the field of causal disentanglement and more controllable generation. The experiments in the original paper was not thorough enough, but the authors have added more analysis in the rebuttal. These additioinal experiments should be included in the final version. Overall, I would recommend acceptance of this paper given its solid contribution.